# Vascular Wall Reactions to Coronary Stents—Clinical Implications for Stent Failure

**DOI:** 10.3390/life11010063

**Published:** 2021-01-17

**Authors:** Tommaso Gori

**Affiliations:** Department of Cardiology, University Medical Center Mainz and DZHK Standort Rhein-Main, 55131 Mainz, Germany; Tommaso.gori@unimedizin-mainz.de; Tel.: +49-6131-172829

**Keywords:** coronary arteries, inflammation, stent, stent thrombosis, stent restenosis

## Abstract

Coronary stents belong to the most commonly implanted devices worldwide. A number of different types of stent exist, with very different mechanical and biochemical characteristics that influence their interactions with vascular tissues. Inappropriate inflammatory reactions are the major cause of the two major complications that follow implantation of stents in a percentage as high as 5–20%. It is therefore important to understand these reactions and how different they are among different generations of stents.

## 1. Introduction

With several million implantations per year, coronary artery stents are among the most commonly used devices for the treatment of a disease that still represents the major cause of mortality and morbidity worldwide [1]. As compared to bare metal stents (BMS), drug eluting stents (DES) have reduced rates of restenosis and target lesion revascularization, profoundly modifying the interventional treatment of coronary artery disease [2]. However, DES failure remains a problem that affects, depending on a number of factors, up to 20% of the devices implanted [1]. In particular, stent thrombosis is a complication associated with a very variable, but not negligible, mortality (from 3.8% in the DESERT (International Drug-Eluting Stent Event Registry of Thrombosis) registry to 45% in the registry by Iakovou et al.) [3,4,5,6,7,8]. While being less malign, stent restenosis is also associated with an acute presentation (unstable angina or myocardial infarction) in about two-thirds of the cases, without difference across stent generations. Our understanding of the pathophysiology of stent failure remains incomplete and these phenomena are surely multifactorial. Patient, procedural and plaque factors obviously play an important role in stent failure [9]; however, it is also clear from observations in both BMS and early-generation DES that the biological reactions induced by the implantation trauma and by the exposure to the stent components are crucial. Far from being only passive supports for the vascular architecture, coronary stents interact with vascular biology and stimulate a number of cellular reactions. Ideally, healing processes should cover the struts with a thin layer of functioning neo-endothelium and minimal or no extracellular matrix that lead to the restitution of vascular function re-establishing laminar flow. Following coronary angioplasty, however, a number of unwanted mechanisms are also activated, as witnessed by systemic increases in c-reactive protein, inflammatory cytokines, and blood cell parameters, all of which are predictors of stent failure during follow-up [10,11,12].

Thus, while the multifactorial nature of stent failure is acknowledged [13], the present review will focus on the biological responses that follow stent implantation, and how these may lead to stent failure across the different generations of stents.

## 2. Stent Complications and Stent Pathology among Different Device Generations

With the constant development of newer device types, the incidence and pattern of their short- and long-term complications have also changed (Figure 1). As compared to balloon-only angioplasty, BMS solved the issue of acute recoil. However, the application of these devices in complex lesions (such as small vessels, bifurcation lesions, diabetic patients, etc.) resulted in a risk of restenosis. Bare metal stent restenosis was characterized by progressive, slow growth of a fibrotic extracellular matrix typically occurring within the first 6–9 months after implantation [2]. While first-generation DES such as Cypher sirolimus-eluting stents (SES, Cordis Corp., Miami Lakes, FL, USA) and Taxus paclitaxel-eluting stents (PES, Boston Scientific, Natick, MA, USA) reduced the incidence of in-stent restenosis, an increase in the incidence of late in-stent thrombosis due to delayed arterial healing and adverse hypersensitivity reactions compromised the safety of these devices [14,15]. Although in-stent thrombosis was not a new phenomenon, its characteristics in patients who had received implantation of BMS where completely different. In BMS, stent thrombosis had typically an earlier onset, and it was almost invariably caused by the hemodynamic changes caused by the growth of neointima [9]. In contrast, in first-generation DES, stent thrombosis occurred later (one or more years after implantation) and it was characterized by evidence of delayed arterial healing, lack of endothelialization, persistent fibrin deposition as well as hypersensitivity reactions with eosinophilic/heterophilic infiltration of the arterial wall as pathognomonic findings [16,17,18,19]. Further, another new clinical entity, in-stent neoatherosclerosis, was described as a long-term consequence of bare metal and particularly first-generation DES: neoatherogenesis represents today one of the most common causes of late and very late stent failure [20]. Its incidence appears to be twice as high in DES-treated segments lesions than in lesions treated with BMS (31% vs. 16%), and it appears to occur earlier after DES than BMS (420 days (361–683 days) versus 2160 (1800–2880) days). As discussed below, late (1 month to 12 months after implantation) and very late (beyond 1 year) [21] stent thrombosis as well as neoatherogenesis appear to be caused (also) by pathological responses to one of the component of the DES.

Second-generation DES were developed explicitly to address these issues of biocompatibility. They feature a thinner strut platform with streamlined strut geometries to decrease prothrombotic fibrin deposition and a thromboresistant polymer as well as different (possibly less toxic) drugs such as everolimus, biolimus, or zotarolimus, which do not inhibit endothelial cell adhesion as compared to tacrolimus [22]. Of importance, while first-generation DES were designed to delay cell proliferation including re-endothelization processes to reduce the risk of restenosis, newer-generation DES were developed to promote a faster re-endothelization through their enhanced biocompatibility. In particular, the introduction of fluoropolymer coatings appears to confer (in a dose-dependent manner) thromboresistance by reducing platelet activation [23,24], an effect that is believed to result from adhesion and retention of albumin on the polymer surface. This albumin-rich protein layer may in turn inhibit the adhesion of prothrombotic proteins such as fibrinogen. A high hydrophobicity, high bond strength, low polarizability and a high degree of fluorination appear to be responsible for these advantages. As well, fluoropolymer-coated everolimus-eluting stents were associated with significantly lower inflammatory cell adhesion as compared to BMS with the same stent platform [25] (Figure 2). Additional features of some modern stents include polymers allowing slow release of the drug (to avoid toxic reactions), drug coating limited to the abluminal side (to avoid impairment of endothelialization), antibodies for the adhesion of endothelial progenitor cells, elution of antioxidant drugs.

Meta-analysis studies confirmed a lower target-lesion revascularization and incidence of stent thrombosis with newer-generation devices as compared to 1st generation DES and BMS [26,27]. The safety profile of second generation DES was also confirmed by intravascular imaging [28], angioscopy and pathology studies confirming improved stent coverage by neoendothelium associated with less aggregates of red blood cells, less peri-strut fibrin deposition, and a lower inflammation score [28,29]. Further evolutions to improve the biocompatibility of implants included the development of antithrombotic polymers (so-called fluoropassivation [25]), bioresorbable polymers (which however do not seem to improve significantly patients outcomes [30]), and completely bioresorbable scaffolds (whose mechanical characteristics, particularly strut thickness however represented a limitation [13,31,32]).

## 3. The Mechanisms of Restenosis

Restenosis is defined as the reduction in lumen size of an artery after intra-arterial intervention. While restenosis prior to the advent of coronary stents was mostly caused by recoil, in-stent restenosis is a response-to-injury process caused by formation of a tissue called neointima. For BMS, the theory postulated that this neointima is mostly composed of fibrotic tissue with a gradual and progressive onset over months, which led to the concept that in-stent restenosis is a benign phenomenon. In contrast, reports on the presentation of restenosis have consistently shown—without differences between BMS and DES—that more than 50% of the patients with in-stent restenosis present with unstable angina and myocardial infarction [33]. The initiating stimulus for restenosis is endothelial denudation and/or mechanical injury to the vessel wall, triggering an inflammatory response which can also be measured in terms of increase in circulatory markers such as c-reactive protein, amyloid A and fibrinogen [34,35]. Upon implantation, endothelial denudation and the exposure of components of the vascular extracellular matrix like collagen causes activation and adhesion of platelets. As well, the stent struts cause, proportionally to their thickness and dependent on the design of the stent, a disturbance in the flow that triggers the formation of large fibrin clots, further causing flow disturbances close to the struts and activating homeostatic reactions characterized by phagocyte invasion. In this context, the immunosuppressive action of the drug eluted might have paradoxically negative effects, inhibiting tissue repair reaction and fibrin removal. These processes might explain the acute presentation of restenosis, as the activation of the thrombotic cascade ensuing from incomplete thrombin removal, although not causing occlusive thrombus, activates the recruitment of inflammatory cells such as monocytes, T cells, neutrophils via expression of adhesion molecules (ICAM, VCAM, intercellular, and vascular adhesion molecules), production of chemoattractant molecules (MCP-1 or monocyte chemoattractant protein-1, Interleukin (IL)-8) by endothelial, and smooth muscle cells and the production of growth factor such as PDGF (platelet-derived growth factor), bFGF (fibroblast growth factor), TGF (transforming growth factor)-beta, IGF (insulin-like growth factor), VEGF (vascular endothelial growth factor, and thrombin (Table 1, Figure 3).

The cytokines involved in these processes include IL-1, IL-6, and TNF-alpha [36]. This cascade is believed to activate a self-sustaining positive feedback loop of autocrine/paracrine messages that leads to stimulation of vascular smooth muscle cells growth and the production of extracellular matrix components. Matrix metalloproteinases activated by plasmin and their tissue inhibitors are also important mediators of this remodeling process. The formation of an endothelium is a critical step in these processes: an efficient endothelium may inhibit smooth muscle cell proliferation and neointimal growth; impaired neoendothelialization is associated with increased neointima proliferation [37]. However, studies have shown that stenting with BMS, and even more with of first generation DES, inhibits endothelialization [38,39]. Several lines of evidence also suggest that the above signaling cascades following coronary damage and stent implantation may also mobilize bone marrow-derived CD34-postive stem cells, which may then differentiate along endothelial or smooth muscle cell lines, in a balance between endothelial-like stem cell responses that favor reendothelialization and smooth muscle-like stem cell responses that promote restenosis [40].

A number of potential therapies have attempted to address or reduce the incidence of restenosis. Based on the concept that oxidative stress may play an important regulatory role in inflammatory states (also supported by the fact that elevated systemic level of oxidative stress markers predict restenosis [41]), a number of trials have tested the effects of antioxidants on the incidence of restenosis. Probucol, a lipid-lowering agent with strong antioxidant properties, has shown to decrease the risk of restenosis and major adverse events by 41% and 31% in a meta-analysis of 15 studies and 859 patients [42]. Probucol-eluting stents have also been tested, but they did not show superiority as compared to newer-generation DES in a large randomized trial [43]. The use of other drugs, such as sirolimus, colchicine, prednisolone and other reviewed in [44] has shown variable results and will need to be further investigated.

## 4. The Mechanisms of Stent Thrombosis

The pathophysiology of in-stent thrombosis is complex and multifactorial, and the vascular responses to stent implantation are only one of the players [9]. An incomplete healing and an impaired endothelization [5] resulting from the toxic effect of stent drug and/or polymer are however important risk factor.

As described above, local and systemic inflammation promote neointimal proliferation, resulting in the pathogenesis of in-stent restenosis. More importantly, chronic low-grade inflammation is also felt to play a primary role in the pathogenesis of stent thrombosis [45,46,47,48]. In particular, as originally observed by the group of Virmani, type IV hypersensitivity, or foreign body-induced activation may be recruited, resulting in a delayed or inefficient stent reendothelialization (Figure 3). These inflammatory processes cause vascular positive remodeling (vessel wall enlargement) and subsequent stent malapposition [47,49], which, in turn, cause flow turbulence and platelet activation [9] (Figure 4).

The histology underlying these phenomena has been investigated in several studies [49]. Eosinophil infiltrates surrounding stent struts may result from a reaction against the metal of the stent, usually 316L stainless steel in the case of BMS and first-generation DES [50]. To this regard, modern stents based on alloys such as platinum chromium or cobalt chromium have a lower nickel content, and other materials, such as zinc-based scaffolds, are being tested with promising results [51]. Further, triggers for hypersensitivity reactions may be the antirestenotic drug and/or the polymer carrying the drug. Polymers in particular have been shown to produce hypersensitivity reactions in humans and porcine models and several case series demonstrating inflammation around a DES have been published [52,53]. These eosinophilic infiltrates have been proposed to result in a specific form arteritis with tissue necrosis and erosion around the stent struts, causing the typical appearance denominated “peri-strut low intensity areas”, positive vessel remodeling with aneurysma and evaginations (Figure 5).

The above processes may also lead to a direct activation of the coagulation cascade and platelet activation. An analysis of thrombus composition in the PRESTIGE cooperation (Prevention of Late Stent Thrombosis by an Interdisciplinary Global European Effort) provides important insights into the pathological processes leading to thrombus formation. In this study, a total of 253 thrombus specimens were analyzed, the majority (68.8%) from cases of late stent thrombosis. Histology showed leukocyte infiltration, neutrophil extracellular traps and eosinophils, with a higher incidence in late stent thromboses of DES [54]. In the RADAR (Research on Adverse Drug Events and Reports) database, DES-induced hypersensitivity reactions were also associated with systemic clinical manifestations suggestive of allergy, including nonurticarial rash, hives, dyspnea, myalgia/arthralgia, itching, and blisters. In a recent imaging study, putative causes of stent thrombosis that could be caused by an inappropriate remodeling process triggered by inflammation were found in 98% of patients with late stent thrombosis. The most frequent findings included strut malapposition (34.5%), neoatherosclerosis (27.6%), uncovered struts (12.1%), and stent under expansion (6.9%) [55].

While the subsequent development of bioresorbable polymers does not seem to have been associated with additional advantages as compared to the antithrombotic fluoropolymer [56,57,58], the progressive reduction of stent strut thickness has led to improved healing and reduced incidence of stent thrombosis [27,59,60,61]. Given the high mortality associated with stent thrombosis, further progress in understanding its mechanisms and its prevention are of primary importance. Medical therapy has a very important impact on the processes described above. Antiplatelet agents are clearly the mainstay in the prevention of thrombosis in the arterial tree. Complementing their effect on platelet aggregation, their anti-inflammatory and endothelium-protective “pleiotropic” effects, as recently demonstrated in the EST (Endothelium, Stent, and Thrombosis) trial and reviewed in [34,45], may have an important additional role. A similar effect of statins has been hypothesized based on observations from the BASKET (Basel Stent Kosten Effektivität Trial) and the Korean Acute Myocardial Infarction Registry [62,63], but remains to be tested in specific trials.

## 5. Neoatherogenesis: A Long-Term Complication

The chronic inflammation and endothelial dysfunction discussed above as causes of restenosis may also induce late de novo neoatherosclerosis. In a recent imaging study, lipid neointima was shown in 40% of the cases of in-stent restenosis, calcific lesions were reported in a similar incidence, and thin-cap fibroatheromas was observed in up to 15%. The frequency of these forms of neoatherosclerosis was significantly greater with DES than BMS (48.4% vs. 23.5%, *p* = 0.018), even though lesions in BMS had a thinner cap [64] (Figure 6).

Backscatter intravascular ultrasound evaluation confirmed that the proportion of restenotic lesions containing lipid pools is larger in DES as compared to BMS, suggesting that the formation of a neoplaque (neoatherosclerosis) may in part be proportionally more important in these devices [65]. Although not specific for DES versus BMS, neoatherosclerosis is therefore proportionally a larger issue for modern stents, which appear to be less at risk for fibrotic restenosis and stent thrombosis. Data from two large registry studies examining the characteristics and possible mechanisms of very late stent failure and comparing early- with new-generation DES confirm that the three most common causes of very late DES failure include strut malapposition (34.5%), neoatherosclerosis (27.6%), and uncovered struts (12.1%) [55]. Importantly, the time pattern of the mechanisms of restenosis appears to be influenced by its pathophysiology: while early (<1 year) in-stent restenosis in second generation DES appears to be (similar to that of BMS) mainly caused by intimal hyperplasia, late (>1 year) restenosis is most frequently caused by neotherosclerosis [66]. Specifically, this latter type of lesions was characterized by higher prevalence of lipid-laden neointima, thick- or thin-cap fibroatheroma, neovessels, microcalcifications, and macrophage pools. Of importance, the mechanisms that lead to the mid-term inhibition of tissue proliferation might actually contribute to the pathophysiology of long-term neoatherosclerosis. An impaired endothelialization leads to decreased nitric oxide production, loose intercellular junctions (with subsequent loss of barrier function) and decreased exposure of antithrombotic barriers, all of which promote infiltration, retention, and increased expression of foamy macrophages within the neointima [67]. As for native atherosclerosis, macrophage cells group to form a fibroatheromatous plaque (“intimal xanthoma”), which can then evolve into a rupture-prone thin-cap fibrous atheroma [68]. Apoptosis of macrophages may also results in the formation of a necrotic core and calcifications may also be present [69]. An impaired endothelialization (and thick stent struts) also cause blood flow turbulence, modifying wall shear stress and local hemodynamic forces and leading to a functionally impaired endothelium [70]. An ineffective endothelial barrier and the expression of adhesion molecules (failing the inhibition of the nitric oxide pathway) results the penetration of inflammatory cells lipoprotein and proteoglycans into the subendothelial space. Most of the drugs released by current DES devices inhibit the mammalian target of rapamycin (mTOR), a phosphatidylinositol 3-kinase-related kinase (PIKK) belonging to the family of serine/threonine protein kinases [71]. mTOR is involved in regulating a number of fundamental cell processes from protein synthesis to autophagy, and the mTORC1 (mTOR complex 1, consisting of mTOR and RAPTOR, regulatory associated protein of mTOR), mammalian LST8/G-protein β-subunit like protein and lethal with sec thirteen 8) regulates the activity of p70, which is vital for cellular proliferation and migration of both endothelial and smooth muscle. By inhibiting p70, mTOR inhibition leads to the prevention of restenosis after injury [72]. However, mTORC1 also promotes cell growth by suppressing protein catabolism, most notably autophagy. As well, mTOR also modulates inflammatory reactions as it blocks T-cell activation, and is also involved in adipogenesis and lipogenesis. Exposure to mTOR inhibitors increases vascular permeability, which might promote neoatherosclerosis [73]. Further, mTORC2 positively regulates Akt activity, whose function includes endothelial survival as well as transcriptional control of VE-cadherin [72]. Translocation of VE cadherin from the membrane to the intracellular space contributes to impaired endothelial barrier function (mechanisms described in [72]). Although these mechanisms still remain incompletely understood, it is however clear that they may depend on ancillary, off-target effects of the antiproliferative drug eluted by the stent. Evidence to this regard is summarized in recent reviews, which also discuss the chances offered by alternative, more specific, drugs [72].

In terms of prevention, statins have been shown in small studies to stabilize neointima and prevent the appearance of neoatherosclerosis [74]. In contrast, although promising, an effect of antioxidants has been proposed but remains to be demonstrated [75].

## 6. Conclusions

Percutaneous coronary interventions cause mechanical injury and vascular inflammation which, along with the presence of a foreign body and the biochemical processes triggered by the polymer and the drug eluted by the stent, cause complex interactions between endothelial cells, smooth muscle cells, platelets, and inflammatory cells including neutrophils, monocytes, and lymphocytes. The activation of signaling cascades leads to repair and vascular healing but also adverse arterial remodeling, neointimal proliferation, and restenosis. Although effectively reduced after the introduction of DES, restenosis, and thrombosis remain important issues with important clinical implications. While the current generations of DES appear to have reached an innovation plateau that allows safe treatment of complex anatomies, future generations of devices, with a better mechanical profile and improved biocompatibility are still awaited.

## Figures and Tables

**Figure 1 life-11-00063-f001:**
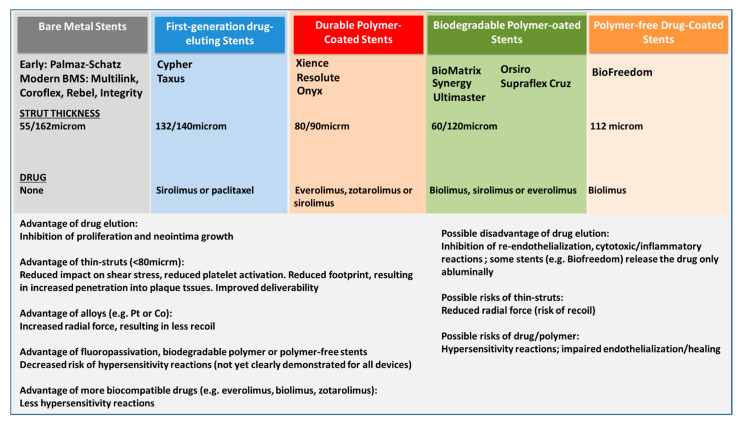
Schematic presentation of the different stent generations, from bare metal stents (BMS) to newer generation drug eluting stents (DES). Only some of the several hundred trademark brands are listed, each device type has individual characteristics, the reader is directed to the manufacturer website or specific publications.

**Figure 2 life-11-00063-f002:**
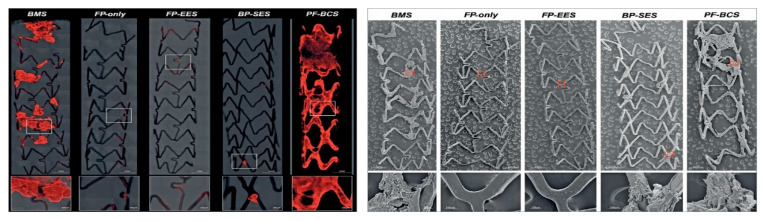
Representative confocal microscopic (**left panels**) and scanning electron microscope images (**right panels**) describing the thrombogenicity of different stent types images of (BMS = bare metal stents; FP: fluoropolymer; FP-EES: fluoro-polymer and everolimus-eluting; BP-SES: bioresorbable polymer with sirolimus; PF-BCS: polymer-free Biolimus A9-coated stents). Left panel: images are obtained using immunofluorescent staining against dual platelet markers (CD61/CD42b) in a swine shunt model. Stents with fluoropolymer (FP-only and FP-EES) were associated with less platelet adhesion. Right panels show evidence of platelet aggregation clot on BMS, BP-SES, and PF-BCS. Image reproduced from [25], with permission.

**Figure 3 life-11-00063-f003:**
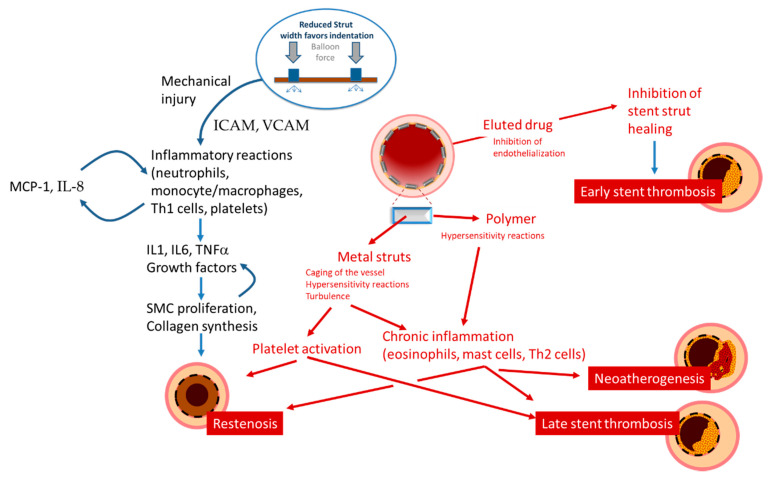
Schematic presentation of the mechanisms leading to restenosis, thrombosis, and neoatherosclerosis.

**Figure 4 life-11-00063-f004:**
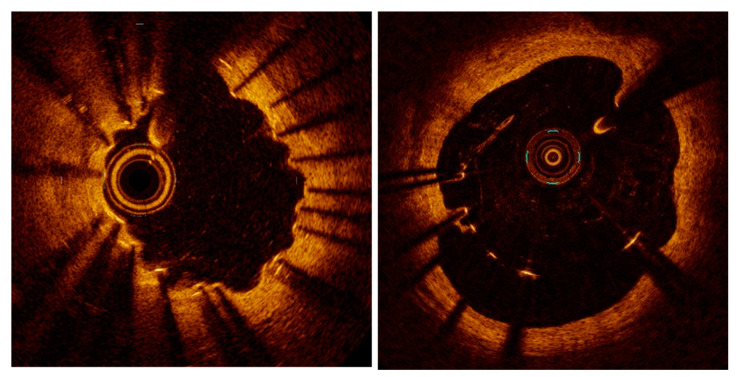
Examples of peri-strut evaginations (**left**) and malapposition (**right**) resulting from positive remodeling (lumen expansion) after stent implantation.

**Figure 5 life-11-00063-f005:**
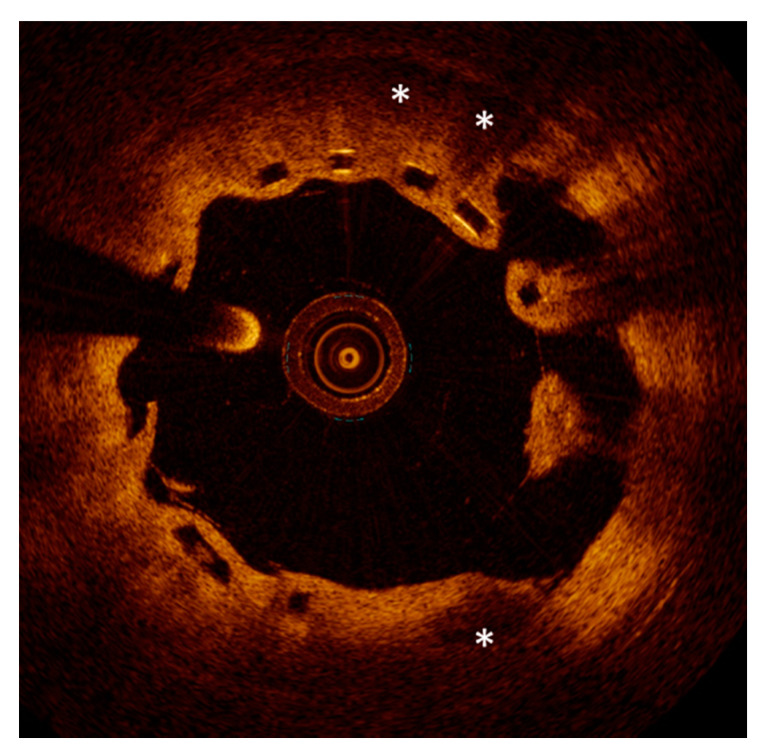
An optical coherence tomography image showing one-year result after implantation of a bioresorbable scaffold. Malapposed struts (3 o’clock), evaginations (8 to 10 o’clock) and peri-strut low-intensity areas (stars) are evident.

**Figure 6 life-11-00063-f006:**
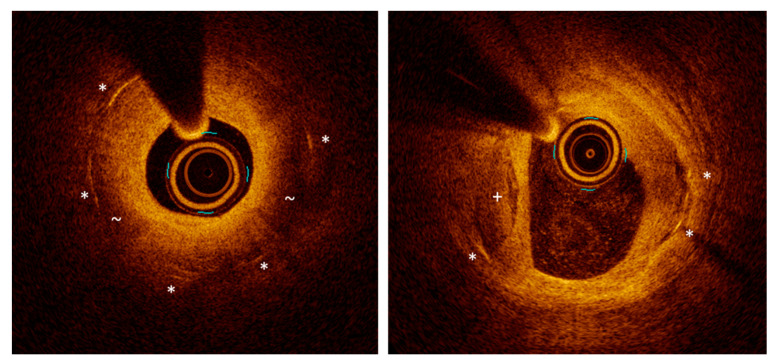
Optical coherence tomography images on neoatherogenesis. (**left**): low-backscattering areas (~) with high attenuation suggestive of atheroma within the stent borders (stent struts are marked with *). (**right**): a calcific plaque (+).

**Table 1 life-11-00063-t001:** Mechanisms and mediators involved in restenosis.

	Mechanism(s) Involved	Mediators and Signaling
**Response to Damage**		
**Proliferation**	Migration of smooth muscle cells, production of membrane metalloproteinases	Cytokines (IL-1, IL-6, TNF-alpha) Growth factors (PDGF, IGF, FGF, VEGF)
**Remodeling**	Remodeling of the neointima, deposition of extracellular matrix, neoatherogenesis	Macrophages/Foam cells Cytokines (IFNgamma) Growth factors (PDGF, TGFbeta, IGF, VEGF)
**Thrombus Formation**	Adhesion and activation of platelets Recruitment and diapedesis of Inflammatory cells	Expression of vWF and TF Turbolent flow Nitric oxide, Thrombin Adhesion molecules Chemotactic factors (IL-8, MCP-1) Cytokines (IL-1, IL-6, TNF-alpha) Growth factors (PDGF, Thrombin)

TNF: tumor necrosis factor; vWF: von Willebrand factor; IFN: interferon; FGF: fibroblast growth factor; all other abbreviations as in text.

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
