# Peer review of "Vascular Wall Reactions to Coronary Stents—Clinical Implications for Stent Failure"

_life, 2021, doi:10.3390/life11010063_

Round 1
Reviewer 1 Report
In this manuscript Gori reviewed the clinical implication of stent failure and the associations to this process for the different generations of stents, with the intention of focusing on the cellular responses that follow stent implantation, and their effects and interactions with the drug eluting stents and vessel wall.
Overall, the paper is very interesting and well written. However, cellular and molecular processes are treated superficially and need further insights and information. Thus, there are major and minor points (detailed below) to be resolved before considering it suitable for publication:
Major Points:
- To be more incisive and clear the author must insert a table with the different stent device generations and their complications and advantages.
- The author must insert a figure, such as a cartoon, summarizing all of the well reported biological mechanisms for restenosis, stent thrombosis and neoatherogenesis.
- Concerning the cellular mechanisms of neoatherogenesis, different biological pathways and targets are reported such as mTOR and PIKK. At the same time, in the conclusion part are reported the processes triggered by the polymer and the drug eluted by the stent, like interactions between endothelial cells, smooth muscle cells, platelets, and inflammatory cells activation. However, these biological processes are addressed too quickly and superficially in the manuscript and need further insights and information.
- Are there any available data on the possible circulating biomarkers to predict the Stent Failure? If yes, the author should insert a small chapter describing these molecules with their predictive power.
Minor Points:
- Some words are disconnected or ill posed, for example in pag. 4 line 169 „allergic, please remove them.
- Some sentences, in the part concerning the mechanisms of stent thrombosis, are not very clear. Please check and explain them better.
- Introduction needs more specific bibliographic citations.
Author Response
Reviewer 1
[…] Overall, the paper is very interesting and well written. However, cellular and molecular processes are treated superficially and need further insights and information. Thus, there are major and minor points (detailed below) to be resolved before considering it suitable for publication
Thank you very much for your constructive comments and suggestions.
Major Points:
- To be more incisive and clear the author must insert a table with the different stent device generations and their complications and advantages.
Thank you very much for your comment. This is a good idea. I have added a new figure (figure 1) in which I tried to summarize the main characteristics of different stent iterations and the advantages/disadvantages of each factor (polymer, drug, strut thickness). The figure summarizes the advantages of modern polymers, drugs, thin-struts, and metal alloys and provides the most common brand names as practical reference. The figure also does not mention features such as open cell design, flexibility, deliverability, shaft characteristics, post-expansion limits which are also important but do not interfere directly with the biological reactions that follow stent implantation.
- The author must insert a figure, such as a cartoon, summarizing all of the well reported biological mechanisms for restenosis, stent thrombosis and neoatherogenesis.
Thank you very much, also a very good idea. I have tried to summarize the mechanisms in Figure 3. This has become a very busy figure, and I had to focus on some of the molecular mechanisms for the sake of clarity. If the reviewer has specific suggestions and corrections, I would of course be happy to improve it.
- Concerning the cellular mechanisms of neoatherogenesis, different biological pathways and targets are reported such as mTOR and PIKK. At the same time, in the conclusion part are reported the processes triggered by the polymer and the drug eluted by the stent, like interactions between endothelial cells, smooth muscle cells, platelets, and inflammatory cells activation. However, these biological processes are addressed too quickly and superficially in the manuscript and need further insights and information.
Thank you very much, this section has been expanded with several additional references and more details on the pathology of neoatherosclerosis.
- Are there any available data on the possible circulating biomarkers to predict the Stent Failure? If yes, the author should insert a small chapter describing these molecules with their predictive power.
Thank you very much. Yes, there is indeed evidence that systemic levels of markers of inflammation are acutely increased after stenting, this is now quoted in the introduction. Also, the evidence that oxidative stress markers predict restenosis is quoted (reference 41). There is also evidence that an increase in these markers may predict the incidence of stent failure. These markers are however not normally used in the clinic, and there are also controversial data. Since the word count of the review is already beyond the maximum allowed, I would therefore prefer to limit this discussion to a mention in the introduction (lines 38-41) instead of adding a whole new paragraph. I hope the reviewer agrees with this decision.
Minor Points:
- Some words are disconnected or ill posed, for example in pag. 4 line 169 „allergic, please remove them.
Thank you very much, I agree with you and thank you for your comment. This has been corrected and the whole paper has been reviewed.
- Some sentences, in the part concerning the mechanisms of stent thrombosis, are not very clear. Please check and explain them better.
Thank you very much, the paragraph has been revised.
- Introduction needs more specific bibliographic citations.
Thank you, you are right. I have added a number of references. The papers reporting the exact incidence of mortality/ACS in patients with thrombosis/restenosis are now quoted.
Reviewer 2 Report
The author of this manuscript presents a review of the vascular wall reactions to coronary stents, and its clinical implications for stent failure.
The topic is of great interest to the medical and scientist community; however, several issues need to be addressed to clarify this manuscript.
Major issues:
- The author mention “by reducing platelet activation” (line80) and “the anti-inflammatory side-effects of antiplatelet agents” (lines 162-163), and there is not given any information about the effect of antiplatelet agents nor statins in the vascular wall after coronary stenting, even neoatherogenesis and lipid pool are well described (5th paragraph).
I would suggest describing and adding some updated information of these commonly used drugs in this clinical setting as this topic is relevant in clinical practice.
- Line 209. I would suggest the definition in front PRESTIGE, mentioning that it is a Consortium that was established in Europe to investigate stent thrombosis.
- Line 214. RADAR database, is it referred to ‘Research on Adverse Drug Events and Reports’? I would suggest the author to define and adding the reference.
- Lines 248-249. Author defines early (<1 year), and late (>1 year) restenosis.
- Very late is not defined (lines 65, 244), the reviewer recommends adding the definition and the references used for its categorization.
Minor issues:
- Bare metal stents, it is commonly written as BMS, I would suggest the author use this form in all manuscript.
- The acronyms in lines 137-140, and at Table 1 as well.
- Author should define all the abbreviations used.
- Line 122. Drug eluting stents should be written DES.
- Lines 193-198. Text is grey, and ‘cobalt’ is underlined.
- Lines 202-203. (Figure 3) is repeated.
- Line 235. ‘areras’, I guess it means ‘areas’
Author Response
Reviewer 2
[…] The topic is of great interest to the medical and scientist community; however, several issues need to be addressed to clarify this manuscript.
Major issues:
The author mention “by reducing platelet activation” (line80) and “the anti-inflammatory side-effects of antiplatelet agents” (lines 162-163), and there is not given any information about the effect of antiplatelet agents nor statins in the vascular wall after coronary stenting, even neoatherogenesis and lipid pool are well described (5th paragraph). I would suggest describing and adding some updated information of these commonly used drugs in this clinical setting as this topic is relevant in clinical practice.
Your comment is well taken, you are right. I have added, at the end of each paragraph („mechanisms or restenosis“, mechanisms of thrombosis“ and „mechanisms of neoatheroclerosis“) a short description of the effect of different drug therapies.
- Line 209. I would suggest the definition in front PRESTIGE, mentioning that it is a Consortium that was established in Europe to investigate stent thrombosis.
You are right, this has been added.
- Line 214. RADAR database, is it referred to ‘Research on Adverse Drug Events and Reports’? I would suggest the author to define and adding the reference.
Thank you, this has been done.
- Lines 248-249. Author defines early (<1 year), and late (>1 year) restenosis.
This has been added (now line 284), thank you.
- Very late is not defined (lines 65, 244), the reviewer recommends adding the definition and the references used for its categorization.
This had been added, thank you.
Minor issues:
- Bare metal stents, it is commonly written as BMS, I would suggest the author use this form in all manuscript.
Changed, thank you.
- The acronyms in lines 137-140, and at Table 1 as well.
You are right, thank you.
- Author should define all the abbreviations used.
Abbreviations have been explained, thank you.
- Line 122. Drug eluting stents should be written DES.
Done, thank you.
- Lines 193-198. Text is grey, and ‘cobalt’ is underlined.
Thank you, corrected.
- Lines 202-203. (Figure 3) is repeated.
Corrected, thank you.
- Line 235. ‘areras’, I guess it means ‘areas’
You are right, thank you.
Round 2
Reviewer 1 Report
After the author's review the manuscript is now acceptable for publication in LIFE
Author Response
Dear Reviewer 1, thank you very much for your positive assessment and your suggestions in the first round.
Reviewer 2 Report
The authors have put a considerable amount of effort in revising this manuscript according to the reviewer’s instructions.
There are three issues at Figure 1:
- I do not see bare metal stents (BMS) in any of the categories. Please clarify.
- The advantages and risk are not clearly seen if they are related to one type of stents or to all.
- The footage of the figure 1 gives information that should be at the main text (‘Additional features of some modern stents include polymers allowing slow release of the drug (to avoid toxic reactions), antibodies for the adhesion of endothelial progenitor cells, elution of antioxidant drugs’)
Author Response
Dear reviewer 2,
1. I apologize for the mistake. The left column was indeed meant to describe BMS, but I did not correct the heading. Thank you for noticing.
2. I have changed the figure and the legend to adress your comments. I made an effort at describing the advantages (or disadvantages) of each feature, rather than each stent brand (the list would be very long, and stents used in Europe differ from those used in USA or Asia). It would be very hard to describe the characteristics of each stent brand, so I opted to list the different general classes (BMS, permanent polymer, thin-struts etc) and the advantages/disadvantages of each feature. Since these are shared in a sparse pattern across the several hundred brands available on the market, one would have to list the advantages/disadvantages of each brand (for instance: biofreedom has abluminal, polymer-free release but its struts are relatively thicker, with 112microm; supraflex cruz has very thin struts with 60microm and biodregradable polymer; xience has fluoropassivation). The list would have to include several hundred entries, and direct clinical comparisons are mostly not available.
3. You are right, this text has been moved to page 4, first paragraph.